# Phytochemical Profile and Bioactive Evaluation of *Porophyllum gracile*

**DOI:** 10.3390/ijms26178350

**Published:** 2025-08-28

**Authors:** María de Guadalupe Ruiz-Almada, Maribel Plascencia-Jatomea, Armando Burgos-Hernández, Hisila del Carmen Santacruz-Ortega, Luis Noguera-Artiaga, Carmen María López-Saiz

**Affiliations:** 1Departamento de Investigación y Posgrado en Alimentos, Universidad de Sonora, Encinas y Rosales s/n, Hermosillo 83000, Sonora, Mexico; a215206757@unison.mx (M.d.G.R.-A.); maribel.plascencia@unison.mx (M.P.-J.); armando.burgos@unison.mx (A.B.-H.); 2Departamento de Investigación en Polímeros y Materiales, Universidad de Sonora, Encinas y Rosales s/n, Hermosillo 83000, Sonora, Mexico; hisila.santacruz@unison.mx; 3Departamento de Tecnología Agroalimentaria, Escuela Politécnica Superior de Orihuela, Universidad Miguel Hernández de Elche, Carretera Beniel, Km 3.2, E-03312 Orihuela, Alicante, Spain; lnoguera@umh.es

**Keywords:** traditional medicine, Asteraceae family, biological activity, antioxidant activity, antiproliferative activity, cytotoxic activity

## Abstract

Plants of the genus *Porophyllum* (*Asteraceae*) have traditional medicinal uses, but only 8 of 25 species have been studied. This study aimed to profile volatile compounds, phenolics, and fatty acids in dried leaves and stems of *Porophyllum gracile* and assess biological activities of extracts obtained using different solvents. GC-MS, HPLC-DAD, and GC-FID analyses identified over 120 compounds, including fatty acids, chlorogenic acid derivatives, quercetin derivatives, terpenes, ketones, aldehydes, and alcohols. Antioxidant activity in vitro (ABTS, DPPH, and FRAP assays) suggested a strong electron-transfer-mediated mechanism. In ARPE-19 cells under doxorubicin-induced oxidative stress, hexane and ethanolic extracts from leaves and stems significantly reduced intracellular reactive oxygen species, in some cases outperforming vitamin E. No antiproliferative activity was detected against cancer cell lines (MDA-MB-231, HeLa, A549, HCT 116, 22Rv1), nor cytotoxicity toward non-cancerous cells (ARPE-19, hFOB 1.19). This first detailed phytochemical characterization of *P. gracile* demonstrates its cellular antioxidant potential and supports its application as a natural antioxidant source in functional foods or nutraceuticals. Future work should elucidate mechanisms, isolate active compounds, and evaluate bioavailability in in vivo models.

## 1. Introduction

Natural products refer to compounds capable of exerting one or more biological activities, obtained from bacteria, fungi, terrestrial and marine plants, and/or marine animals [1]. Bioactive compounds represent an important source of diverse and largely complex molecules, capable of addressing human needs and challenges [2]. Biologically active compounds primarily belong to secondary metabolism of organisms, which provide them with the necessary tools for adaptation and survival in their native environments; this is why these molecules exhibit unique characteristics and properties [3,4]. Although few natural products are approved for medical use or direct human consumption, many serve as models for synthesis of more effective compounds—those with higher bioavailability [5,6].

Genus *Porophyllum* belongs to the Asteraceae family, which includes plants known for their diverse and high content of bioactive compounds, as well as their use in food production [7]. Species of genus *Porophyllum* are native to the Western Hemisphere, ranging from southern United States to South America, and thrive in tropical and subtropical areas [8,9]. Although there are some reports regarding the bioactive capacity and presence of natural products in plants of genus *Porophyllum* [8,10,11,12,13,14,15,16,17], studies on bioactive properties of *P. gracile* are scarce [18], despite its long-time use in traditional medicine by Sonoran ethnic groups in Mexico [19].

The use of traditional medicine based on plants still dominates many therapeutic practices worldwide. Bioactive compounds in plants remain essential for the development of modern medicine and represent a substantial portion of current pharmaceutical agents [4].

Besides what was reported by Guillet et al. [18], there are no studies that have evaluated phytochemical composition and bioactive compounds in *P. gracile*, which creates a gap in the existing knowledge about this species. Therefore, a detailed characterization of phytochemical composition of *P. gracile* will not only describe the compounds present, but also help identify previously unreported compounds, bioactive compounds, and those with potential future applications.

Due to the traditional medicinal use of *P. gracile* and considering the studies reported for other species of the same genus, it has been hypothesized that *P. gracile* contains a high diversity of molecules with different chemical natures and high bioactive potential, specifically in vitro antioxidant effects and antiproliferative effects on cancer cells. Therefore, the aim of this study was to determine the profile of volatile compounds, phenolic compounds, and fatty acids in the dried leaves and stems of *P. gracile*, as well as to explore the antioxidant (in vitro and in cells), antiproliferative (in cancer cells), and cytoprotective (in apparently healthy cells) effects of *P. gracile* extracts obtained using polar and non-polar solvents.

## 2. Results

### 2.1. Analysis of Volatile Compounds

Chromatograms obtained by GC-MS for the identification of a volatile compound profile in the *P. gracile* plant are shown in Figure A1. The volatile compounds identified in leaves are presented in Table 1, and those identified in stems are shown in Table 2. In this study, 62 compounds were identified in leaves and 46 in stems, using dried and ground samples of *P. gracile*. Among the volatile compounds identified, β-myrcene and caryophyllene were the most abundant in leaves, compared to the other compounds detected, whereas in stems, caryophyllene, α-isomethyl ionone, and kessane were the most prevalent. In both leaves and stems, compounds of diverse chemical and structural nature were identified, including terpenoids (monoterpenes and sesquiterpenes), as well as aldehydes, ketones, alcohols, alkenes, epoxides, ethers, carboxylic acids, and phenols.

### 2.2. Subsection Identification of Phenolic Compounds

Chromatograms obtained by HPLC-PDA-ESI-MSn and phenolic compounds identified in leaves and stems of *P. gracile* are presented in Figure 1. After analyzing the results, it was possible to identify the presence of seven compounds in leaves: 5-O-caffeoylquinic acid (chlorogenic acid), 3-O-caffeoyl-2-C-methyl-D-erythronic acid, 2-O-caffeoyl-2-C-methyl-D-erythronic acid, quercetin pentoside hexoside, quercetin hexoside, quercetin pentoside, and quercetin rhamnoside. Additionally, six phenolic compounds were identified in stems: 5-O-caffeoylquinic acid, 3,5-O-dicaffeoylquinic acid, 4,5-O-dicaffeoylquinic acid, quercetin pentoside hexoside, quercetin hexoside, and quercetin pentoside. Furthermore, presence of 3-O-caffeoyl-2-C-methyl-D-erythronic acid or 2-O-caffeoyl-2-C-methyl-D-erythronic acid was also observed in stems; however, it was not possible to confirm which of these two compounds is present due to similarity in their structures.

### 2.3. Identification of Fatty Acids

Fatty acid profile of leaves and stems of *P. gracile* is presented in Table 3. In this study, it was possible to identify the presence of 19 fatty acids in leaves of *P. gracile*, as well as confirm the absence of 36 other fatty acids in them. C18:3 c9,12,15 alpha/C18:3 n3 (alpha-linolenic acid) was identified as the most concentrated fatty acid in this anatomical region of the plant. On the other hand, regarding stems of *P. gracile*, the presence of 18 fatty acids was identified, and absence of 37 others was confirmed, with C18:2 c9,12/C18:2 n6 (linolelaidic acid) being the most abundant in this anatomical region.

### 2.4. Extraction of Crude Extracts

Six crude extracts were obtained from the extraction process using three solvents (hexane, ethanol, and water) and two anatomical regions of plant (leaves and stems). The yields obtained are shown in Table 4, where the highest yields were observed in aqueous and ethanolic extracts of leaves.

### 2.5. Antioxidant Activity

Results of antioxidant activity evaluation using the ABTS method are presented in Figure 2. It is evident that all extracts, at the highest tested concentrations achieved 100% radical inhibition, while at lower concentrations, inhibition exceeded 50% demonstrating a significant dose–response effect for all six extracts.

The observed dose–response effects allowed the determination of mean inhibitory concentrations (IC_50_) of the ABTS radical for each extract, which are shown in Table 5.

According to the literature, IC_50_ values are used to classify the antioxidant potential of extracts as follows: very potent (IC_50_ < 50 μg/mL), strong or active (IC_50_ between 50 and 100 μg/mL), moderate (IC_50_ between 101 and 250 μg/mL), weak (IC_50_ between 251 and 500 μg/mL), and inactive (IC_50_ > 500 μg/mL) [20,21]. Thus, hexanic extract from leaves of *P. gracile* is classified as active exhibiting strong antioxidant activity. In contrast, hexanic extract from stems, ethanolic extract from both leaves and stems, and aqueous extracts from both leaves and stems are categorized as very potent antioxidant extracts, with IC_50_ values below 50 µg/mL.

Results of antioxidant activity evaluation using the DPPH radical are presented in Figure 3. Although dose–response effects were observed, even at the highest concentration tested, considered extremely high according to the classification mentioned earlier, most of extracts inhibited less than 50% of the radical. Consequently, it was not possible to determine the IC_50_ values for these extracts, as concentrations exceeding 500 µg/mL would be required. Based on established classification, all six extracts are considered inactive against the DPPH radical.

Results of antioxidant activity evaluation using the FRAP method are presented in Table 6. It can be observed that ethanolic extract of leaves, ethanolic extract of stems, and aqueous extract of leaves exhibited the highest activity compared to other extracts evaluated, with ethanolic extract of leaves being the most active.

### 2.6. Cell Viability Assay

Results for antiproliferative activity evaluation in five cancer cell lines are shown in Figure 4. All cell lines exhibited similar responses to evaluated extracts, with none of the six *P. gracile* extracts affecting cancer cell proliferation. No dose–response effects were observed, and differences were not statistically significant. Absence of antiproliferative activity becomes even more evident when compared to the pronounced effect of the positive controls, doxorubicin and cisplatin.

Conversely, results for proliferation in two non-cancerous cell lines, ARPE-19 (retinal pigment epithelial cells) and hFOB 1.19 (osteoblasts), are presented in Figure 5. Evaluated extracts showed no impact on viability of non-cancerous cells.

### 2.7. Cellular Antioxidant Activity Assay

The results for the relative fluorescence intensity in the evaluation of cellular antioxidant activity using DCFH-DA are presented in Table 7. The hexanic extracts from both leaves and stems, and the ethanolic extract from stems, at the tested concentrations (100 µg/mL) were able to reduce oxidative stress in apparently healthy ARPE-19 cells subjected to doxorubicin-induced stress. Relative fluorescence intensity values were lower than those of the control without protective stimulus and even lower than those of the positive control, in which vitamin E was used as the antioxidant agent. These findings are consistent with the fluorescence microscopy images shown in Figure 6, where the intensity of the fluorescence signal decreased in cells treated with *P. gracile* extracts compared with the stressed control.

## 3. Discussion

Plants of genus *Porophyllum* belong to the Asteraceae family, which have been used in traditional medicine throughout history, and there are various reports regarding their bioactive components [9]. However, specifically within genus *Porophyllum*, only eight species have been investigated in terms of their phytochemical composition or evaluation of their bioactive properties, out of a total of 25 species comprising the genus [8,9,11,12,13,17,18,22,23,24,25,26,27,28]. It has been reported that plants of this genus contain terpenes, aldehydes and alcohols, alkenes and alkanes, furan and thiophene derivatives, certain aromatic ethers, thiols and fatty acids, among other compounds.

However, specifically for the species *Porophyllum gracile*, only identification of seven compounds has been reported: α-pinene, sabinene, myrcene, β-cubebene, 7-tetradecene, pentadecanal, and heptadecanal [18]. Therefore, the results obtained in the present study represent a significant contribution to the knowledge of the phytochemical composition of *P. gracile*, as they allowed the identification of more than 120 distinct compounds in leaves and stems. This phytochemical diversity, previously unreported for the species, includes mainly fatty acids, chlorogenic acid and its derivatives, quercetin-derived compounds, terpenes, aldehydes, ketones, and alcohols, among others. These findings substantially expand the known phytochemical profile of *P. gracile* and provide a solid foundation for future research on its biological and pharmacological potential.

It is important to highlight that, although some of compounds reported in the present study coincide with those previously identified in other species of the genus *Porophyllum* [9], the detection of chlorogenic acid derivatives, quercetin-derived compounds, and certain fatty acids represent novel findings within the genus. This reinforces the uniqueness of the phytochemical profile of *P. gracile* and highlights its potential as a source of bioactive metabolites.

On the other hand, regarding evaluated antioxidant activity, ABTS and DPPH radical-based assays are considered mixed-mode methods capable of assessing antioxidant activity through both single electron transfer (SET) and hydrogen atom transfer (HAT) mechanisms. However, the specific information they provide depends on reaction conditions. Rapid decolorization reactions of ABTS reagent indicate radical stabilization via electron transfer [29] as observed in the present study, suggesting the possible existence of an electron transfer-based antioxidant mechanism by compounds present in extracts. This hypothesis is supported by the limited activity observed against DPPH radical, an assay that primarily evaluates hydrogen atom transfer mechanisms in the conditions evaluated [29]. Further support for this proposed mechanism comes from FRAP assay results, as the extracts exhibiting the highest activity in this method (ethanolic leaf extract, ethanolic stem extract, and aqueous leaf extract) also demonstrated the greatest antioxidant capacities against ABTS radical.

Conde-Hernández & Guerrero-Beltrán [17] reported antioxidant activity against ABTS radical from extracts of the plant *Porophyllum ruderale*, obtaining the highest activity from ethanol–water extracts compared to aqueous or pure ethanol extracts. They also observed better results when using fresh plant material rather than dried material as, according to authors, some antioxidant molecules are sensitive to heat, while others may evaporate during the drying process. It is important to note that in the present study, no extractions were performed with solvent mixtures, as this is the first analysis of this nature using *P. gracile*, so the possibility of obtaining higher activity using ethanol–water for serial extraction, as well as using fresh plant material, cannot be ruled out. On the other hand, Jiménez et al. [25] reported high antioxidant activity from *Porophyllum tagetoides* extracts against the DPPH radical, which contrasts with the findings of the present study where extracts obtained from *P. gracile* did not show activity against this radical according to the established limits. In both studies, authors attributed the antioxidant activity to presence of phenolic compounds in plants; however, in present research, although phenolic compounds were detected, their concentration is likely to have been lower compared to other compounds present in extracts, which may have contributed to reduced antioxidant activity observed against DPPH.

Furthermore, regarding antioxidant activity observed and compounds identified in the present study, chlorogenic acid has been reported to exert various bioactive properties related to its ability to regulate inflammatory and oxidative processes, as well as to maintain metabolic homeostasis by effectively eliminating excess free radicals [30]. Quercetin, for its part, is a flavonoid that has been reported as a potent antioxidant, as well as having anti-inflammatory, antiproliferative, anticancer, antidiabetic, antiviral properties, and protective effects against neurodegenerative diseases. In food sources, quercetin is typically found in the form of quercetin glycosides, such as those identified in *P. gracile* in the present study [31,32,33]. Regarding fatty acids, α-linolenic acid, a major fatty acid found in leaves and one of the main components in stems in this study, is an essential omega-3 fatty acid which has already been described for its ability to counteract oxidative stress by decreasing the concentration of ROS, reducing lipid peroxidation, and increasing levels of antioxidant enzymes such as superoxide dismutase, glutathione peroxidase, and catalase, besides being the precursor of EPA and DHA [34,35]. In addition, it has been associated with reduced incidence of stroke and coronary heart disease, as well as antitumor properties, among others [36]. Moreover, the presence of several terpenes was detected; these compounds are main constituents of essential oils and have been widely reported to exhibit various bioactivities, including antioxidant, antiplasmodial, antiviral, anticancer, antidiabetic, anti-inflammatory, and antiseptic properties, among others [37].

Regarding the evaluation of cell viability across different human cancer cell lines, it is important to highlight that one of the most crucial controls when analyzing effects on cancer cells is the use of non-cancerous cells. This is because, in search for potential anticancer drugs, the aim is for these compounds to be specific and to primarily target malignant tissues. However, such specificity remains one of the main challenges of current treatments, which often results in side effects and long-term consequences for patients [38,39].

In this case, evaluated extracts did not significantly affect the apparently healthy cells, which is a desirable outcome. However, as previously mentioned, they also showed no effect on cancer cells, indicating that at the concentrations used in this study, they do not possess antiproliferative activity. According to the United States National Cancer Institute, an extract is considered to have antiproliferative activity if its IC_50_ value is below 30 µg/mL [40]; therefore, it can be confirmed that analyzed extracts do not exhibit activity against cancer cell lines included in this study.

Nevertheless, it cannot be ruled out that extracts may contain individual compounds with the ability to reduce cell viability, but which are present at very low concentrations, potentially masking their effect within the complex mixture of the extract. Likewise, the presence of antagonistic compounds that interfere with or counteract activity of those with antiproliferative potential cannot be excluded, thereby preventing the desired effects on cancer cells from being observed.

The DCFH-DA probe is a non-fluorescent, lipophilic compound that, upon entering the cytoplasm, is hydrolyzed by intracellular esterases to 2′,7′-dichlorodihydrofluorescein (DCFH) through the removal of acetate groups. DCFH is subsequently oxidized by intracellular ROS to form 2′,7′-dichlorofluorescein (DCF), a highly fluorescent molecule; therefore, the intracellular ROS concentration is directly proportional to the observed fluorescence intensity [41,42].

In this study, ARPE-19 cells exposed to the hexanic extract from leaves, hexanic extract from stems, and ethanolic extract from stems exhibited lower relative fluorescence intensity compared to the control without extracts. Notably, these extracts produced a fluorescence intensity even lower than that produced by the positive control (vitamin E). Although all six *P. gracile* extracts demonstrated in vitro antioxidant activity, primarily through electron transfer as evidenced by the ABTS and FRAP assays, only the three above-mentioned extracts exerted this effect within the cells. This outcome may be attributed to the chemical nature and polarity of the compounds present in the hexanic and ethanolic extracts, which typically include mainly lipophilic or moderate polar molecules, consistent with the nature of these solvents, and are more capable of permeating into the cell membrane and exerting antioxidant effects intracellularly [43]. In contrast, the polar compounds present in the aqueous extracts likely had limited membrane permeability, preventing them from displaying antioxidant activity in the cellular assay [44].

Although *P. gracile* has been identified as containing compounds with highly promising antioxidant activity, based on these results, it is recommended to evaluate its capacity to scavenge reactive nitrogen species (RNS) as well as to assess its activity in vivo and potential anti-inflammatory effects. Furthermore, it is essential to conduct bioactivity-guided purification and isolation studies to identify the compounds responsible for the observed bioactivity and to elucidate underlying mechanisms of action.

In addition, *P. gracile* is commonly used in traditional medicine in the state of Sonora, Mexico, to treat circulation-related disorders, such as varicose vein formation. Therefore, it is advisable to investigate its bioactive potential in this context, including its possible antihypertensive activity, for instance, through inhibition of key enzymes such as angiotensin-converting enzyme (ACE).

## 4. Materials and Methods

### 4.1. Plant Material

*Porophyllum gracile* was collected from a single location, specifically the Port of Yavaros, Huatabampo, Sonora, Mexico (26°42′27.3″ N, 109°33′00.4″ W), during October 2023 and October 2024. The plant material was taxonomically identified at the Herbarium of the Universidad de Sonora (Catalog#:31731, Occurrence ID: 6bfc9716-c0a3-47d7-9617-a0e6582bfc6b). For each collection season, approximately 10 kg of plant material was harvested. Samples were dried at room temperature in the dark for three days, and leaves and stems were manually separated and milled using a coffee grinder mill. Each milled sample was homogenized to ensure uniformity, and representative portions were taken, as appropriate, for each of the subsequent analyses.

### 4.2. Analysis of Volatile Compounds

Extraction of volatile compounds was carried out using headspace solid-phase microextraction (HS-SPME) [45]. A 1 g sample (crushed leaves and stems) was placed in 20 mL vials with polypropylene caps and PTFE/silicone septa. Extraction was performed using an automatic sampler AOC-6000 Plus (Shimadzu, Kyoto, Japan), employing a DVB/CAR/PDMS fiber (50/30 μm, 1 cm) at 40 °C for 40 min. Analysis was performed on a gas chromatograph GC2030 (Shimadzu, Canby, OR, USA) coupled with a triple quadrupole mass spectrometer TQ8040 NX, with a Sapiens X5MS column (30 m × 0.25 mm × 0.25 μm, Teknokroma, Cugat del Vallès, Spain). The Q3 Scan acquisition mode (5000 uma/s, 35–400 m/z) was used, with the following oven temperature program: 80 °C, increasing by 3 °C/min to 210 °C, then 20 °C/min to 280 °C (2 min). Helium was used as carrier gas (1.0 mL/min), with a split injection (1:25) and pressure of 65.2 kPa. The injector, ion source, and interface were adjusted to 230, 230, and 280 °C, respectively. Compound identification was based on three criteria: retention index, comparison with retention times of pure standards, and mass spectrum analysis using the National Institute of Standards and Technology (NIST) database. Only fully identified compounds are reported. The analyses were performed in duplicate.

### 4.3. Identification of Phenolic Compounds

Analysis was performed on an HPLC Agilent 1200 Series (Agilent Technologies, Santa Clara, CA, USA); equipped with a photodiode array detector (G1315D), a tandem mass spectrometer, a binary pump (G1376A), a degasser (G1379B), and an automatic sampler (G1313A), using Nucleosil^®^ 100-5 C18 column (25.0 cm × 0.46 cm, 5 µm; Macherey-Nagel, Düren, Germany). A 20 µL sample was injected in triplicate. Mobile phase consisted of a mixture of water and formic acid (99:1, *v*/*v*) as eluent A, and acetonitrile as eluent B, with the following gradient: 10% B (start), 24% (24 min), 32% (28 min), 40% (36 min), 60% (40 min), and 95% (42 min), followed by a 5 min wash and a return to initial conditions. Mass detector conditions were capillary temperature of 350 °C, capillary voltage of 4 kV, nebulizer pressure of 65 psi, nitrogen flow rate of 11 L/min, and mobile phase flow rate of 0.8 mL/min. Helium was used as collision gas in an ion trap with fragmentation voltages ranging from 0.3 to 2 V [46]. Data acquisition was performed in negative ionization mode and automated MSn for the most abundant fragment ion. The ChemStation for LC 3D Systems Rev. B.01.03SR2 software (Agilent, Madrid, Spain) controlled equipment was used. Identification of phenolic compounds was based on order of elution, retention time, and UV-Vis and mass spectra, compared with authentic standards and previous data.

### 4.4. Identification of Fatty Acids

For derivatization of fatty acids [45], 25 mg of a sample were mixed with 2 mL of sodium methoxide in test tubes with caps and incubated in an orbital bath at 50 °C for 10 min. The mixture was then rapidly cooled to 4 °C and 3 mL of 10% acetylene chloride in methanol were added, followed by a 50 °C bath with stirring for 10 min. After reaching room temperature, 7.5 mL of 6% K_2_CO_3_ and 1 mL of internal standard C13:0 (1 mg/mL in hexane) were added, and the mixture was vortexed for 30 s. Samples were centrifuged at 200× *g* at 4 °C for 10 min. The organic phase was transferred to Eppendorf tubes, and a second centrifugation was performed at 7500× *g* at 4 °C for 10 min. Finally, the organic phase was deposited into amber vials for chromatography. Analysis was carried out on a gas chromatograph (Shimadzu Nexis GC-2030; Shimadzu Corporation, Kyoto, Japan) with a flame ionization detector and an automatic injector (AOC-20i) [47]. Helium was used as carrier gas (flow rate 1.05 L/min) and a CP-Sil 88 capillary column (100 m, 0.25 mm internal diameter, 0.20 μm thickness; Agilent Technologies, Santa Clara, CA, USA) was employed. Injector and detector temperatures were set to 250 and 260 °C, respectively, with a split ratio of 1:20. Oven temperature was programmed as follows: 45 °C (4 min), increasing by 13 °C/min to 175 °C (27 min), followed by a 4 °C/min increase to 215 °C (35 min). Fatty acid methyl esters were identified by comparing retention times with the Supelco 37 Component FAME MIX (Merck KGaA, Darmstadt, Germany). Results were expressed as a percentage of each fatty acid in the total profile and as mg/kg of a sample. The analyses were performed in duplicate.

### 4.5. Extraction of Crude Extracts

Milled samples of leaves and stems were separately extracted in darkness using a 1:9 ratio (*w/v*) of different solvents, sequentially from low to high polarity (hexane, ethanol and water), for 24 h in each under constant agitation at 25 °C. Extracts were filtered and dried under reduced pressure using a rotary evaporator (Yamato, RE300, Tokyo, Japan) at no more than 45 °C, then suspended in their respective extraction solvents at a known concentration. Finally, extracts were stored at −20 °C in amber glass vials until further use [48,49,50].

### 4.6. Antioxidant Activity

#### 4.6.1. 2,2′-Azino-Bis(3-Ethylbenzothiazoline-6-Sulfonic Acid) (ABTS•+) Radical Assay

Antioxidant activity was assessed using the assay described by Re et al. [51]. The ABTS reagent was prepared by mixing ABTS solution (3.84 mg/mL) and potassium persulfate solution (0.66 mg/mL) in a 1:1 ratio (*v*/*v*), followed by incubation in the dark at room temperature for 24 h. ABTS solution was adjusted to an absorbance of 0.7 ± 0.01 at 713 nm. Then, 900 µL of ABTS solution was mixed with 100 µL of extracts to achieve concentrations of 31.25, 62.5, 125, 250, and 500 µg/mL. The mixture was incubated for 30 min at room temperature, and absorbance was measured at 713 nm using a microplate reader (Agilent 800 TS; Agilent Technologies, Santa Clara, CA, USA). Trolox was used as a positive control. Three independent assays with five replicates were performed. Results were expressed as a percentage of radical inhibition using the following formula:% of radical inhibition = [1 − [ABS_sample_ − ABS_neg_._control_)/(ABS_ABTS_ − ABS_solvent_)]] × 100(1)
where

ABS_sample_: ABTS + Sample Absorbance.

ABS_neg.control_: Sample Absorbance.

ABS_ABTS_: ABTS Absorbance.

ABS_solvent_: Solvent Absorbance.

#### 4.6.2. 2,2-Diphenyl-1-Picrylhydrazyl (DPPH•) Radical Assay

Antioxidant activity was assessed following the methodology described by Osuna-Ruiz et al. [50]. A DPPH solution was prepared by dissolving reagent in the assay solvent until an absorbance value of 0.7 ± 0.01 was obtained at a wavelength of 520 nm. Subsequently, 900 µL of the DPPH reagent was mixed with 100 µL of extracts at final concentrations of 31.25, 62.5, 125, 250, and 500 µg/mL. The mixture was then incubated for 30 min in the dark at room temperature. Absorbance was measured at 520 nm using a microplate reader (Agilent 800 TS; Agilent Technologies, Santa Clara, CA, USA). Trolox was used as a positive control. Three independent assays with five replicates were performed. Results were expressed as a percentage of radical inhibition using the following formula:% of radical inhibition = [1 − [ABS_sample_ − ABS_neg_._control_)/(ABS_DPPH_ − ABS_solvent_)]] × 100(2)
where

ABS_sample_: ABTS + Sample Absorbance.

ABS_neg.control_: Sample Absorbance.

ABS_DPPH_: DPPH Absorbance.

ABS_solvent_: Solvent Absorbance

#### 4.6.3. Ferric Reducing Antioxidant Power (FRAP) Assay

The FRAP reagent was prepared by mixing 10 mM TPTZ (2,4,6-Tris(2-pyridyl)-s-triazine) in 40 mM HCl, 20 mM FeCl_3_ in ultrapure water, and CH_3_COONa buffer (pH 3.6, 1.55 g/L) in a 1:1:10 ratio. For the assay, 990 µL of FRAP reagent was mixed with 10 µL of the extracts at concentrations of 31.25 or 62.5 µg/mL, followed by incubation at room temperature for 10 min. Absorbance was measured at 593 nm using a UV-Vis spectrophotometer (Uvikon XS, Bio-Tek Instruments, Saint Quentin Yvelines, France) [52]. One assay with three replicates was performed and results were expressed as Trolox equivalents (mmol TE/g of the initial sample).

### 4.7. Cell Culture

For cell viability assays, five cancer cell lines were used: MDA-MB-231 from invasive breast adenocarcinoma, HeLa from cervical carcinoma, A549 from lung carcinoma, HCT 116 from colon carcinoma, and 22Rv1 from prostate carcinoma. Two apparently healthy cell lines were also used: ARPE-19 from retinal epithelium and hFOB 1.19 from osteoblasts (ATCC, Rockville, MD, USA). Cells were cultured in growth medium using Roswell Park Memorial Institute (RPMI) medium (Sigma-Aldrich, St. Louis, MO, USA) for 22Rv1 cell line and Dulbecco’s Modified Eagle Medium (DMEM) (Sigma-Aldrich, St. Louis, MO, USA) for the other cell lines; both mediums were supplemented with 10% fetal bovine serum (Gibco^®^, Thermo Fisher Scientific, Waltham, MA, USA) and maintained in a Binder incubator (BINDER GmbH, Tuttlingen, Germany) at 37 °C, with a 5% CO_2_ atmosphere and 80% humidity.

### 4.8. Cell Viability Assay

The effect of *P. gracile* extracts on cell viability was evaluated using the MTT assay (3-(4,5-dimethylthiazol-2-yl)-2,5-diphenyltetrazolium bromide) [53]. Briefly, 10,000 cells were seeded per well in a 96-well microplate; after 24 h of incubation at 37 °C, culture medium was replaced with treatments at concentrations of 50 and 100 μg/mL in culture medium. Cells were then incubated for 48 h; after this period, 10 μL of MTT solution (5 mg/mL in water) was added, and cells were incubated for 4 h. Finally, the medium was removed and the MTT formazan crystals were dissolved in DMSO for absorbance readings at 570 nm, using 630 nm as reference (Agilent 800 TS microplate reader). Cisplatin and doxorubicin were used as positive controls. Three independent assays with three replicates were performed.

### 4.9. Cellular Antioxidant Activity Assay

The cellular antioxidant potential of the extracts was evaluated using a fluorescence intensity assay with 2′,7′-dichlorodihydrofluorescein diacetate (DCFH-DA), following the methodology described by Kim and Xue [41] with slight modifications. Briefly, ARPE-19 cells were seeded in 96-well microplates (10,000 cells/well) and incubated for 24 h at 37 °C. The culture medium was then replaced with 50 µL of the test extracts diluted in medium to a final concentration of 100 µg/mL. After 24 h of incubation with the treatments, 50 µL of doxorubicin solution (4 µM in culture medium) was added to induce ROS generation. Following an additional 24 h incubation, the medium was removed, and the cells were washed with 100 µL of PBS. Cells were then stained with 100 µL of working solution (10 µM DCFH-DA in culture medium) and incubated for 30 min at 37 °C. The staining solution was discarded, and the wells were washed twice with 50 µL of PBS before adding 100 µL of PBS for subsequent fluorescence intensity measurement in a microplate fluorescence reader FLUOstar Omega (BMG LABTECH GmbH, Ortenberg, Germany) at an excitation wavelength of 485 nm and an emission wavelength of 520 nm, with gain set to 1200. Vitamin E (0.4 mM) was used as a positive control [54]. All assays were performed in triplicate. Data were expressed as relative fluorescence intensity (I/I_o_) according to the following equation:(3)$$\frac{I}{I_{o}}=\frac{{{Fl}_{Cells+Tx+Doxo}-Fl}_{Tx+Doxo}}{{Fl}_{Cells+Doxo}}$$
where

Fl_Cells+Tx+Doxo_: Fluorescence intensity of cells treated with extract plus doxorubicin.

Fl_Tx+Doxo_: Fluorescence intensity of extract plus doxorubicin, without cells.

Fl_Cells+Doxo_: Fluorescence intensity of cells treated with doxorubicin alone.

### 4.10. Experimental Design

For the phytochemical characterization, descriptive statistics were used. For the evaluation of biological activities, a completely randomized experimental design was employed.

### 4.11. Statistical Analysis

For analysis of the obtained data, a significance level of 95% (α < 0.05) was established and the JMP 9 statistical package was used. A Probit regression model was employed to determine median inhibitory concentrations (IC_50_). Additionally, a one-way analysis of variance (ANOVA) was performed followed by Tukey’s multiple comparison test.

## 5. Conclusions

The results presented provide a basis for characterizing the compounds present in *Porophyllum gracile*, demonstrating that this species contains a diversity of molecules with varied structural characteristics, including bioactive compounds with promising functional properties. Regarding the bioactive properties evaluated, it can be concluded that the tested extracts possess high in vitro antioxidant activity, primarily through a mechanism mediated by electron transfer. Notably, the hexane and ethanolic extracts from both leaves and stems exhibited a marked antioxidant effect in apparently healthy retinal epithelial cells (ARPE-19), significantly reducing oxidative stress generated by reactive oxygen species. However, these extracts did not show antiproliferative activity against the cancer cell lines tested, and they did not affect the viability of the non-cancerous cell lines evaluated. To the best of our knowledge, this is the first study to provide a detailed phytochemical characterization of *P. gracile*, establishing a basis for future research aimed at evaluating its biological activities, isolating and identifying active compounds, elucidating mechanisms of action, and exploring its potential in therapeutic or nutritional applications.

## Figures and Tables

**Figure 1 ijms-26-08350-f001:**
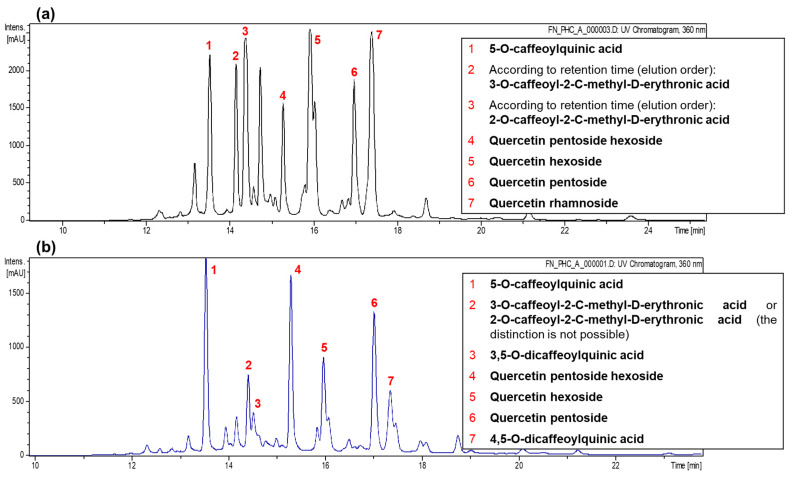
Chromatograms recorded at 360 nm using HPLC-PDA-ESI-MS^n^ and phenolic compounds identified from leaves (**a**) and stems (**b**) on a dry weight basis of *P. gracile* plant, based on two independent analyses.

**Figure 2 ijms-26-08350-f002:**
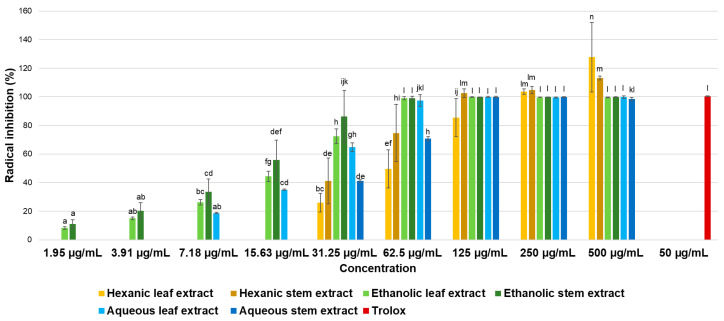
Antioxidant activity of hexane, ethanol, and water extracts from leaves and stems of *P. gracile* determined by the ABTS method. Data are expressed as mean ± SD from three independent assays with five replicates each. Different letters indicate a significant difference (α < 0.05, Tukey’s multiple comparison test).

**Figure 3 ijms-26-08350-f003:**
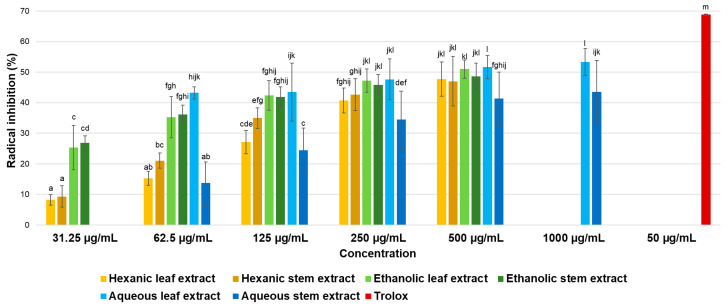
Antioxidant activity of hexane, ethanol, and water extracts from leaves and stems of *P. gracile* determined by the DPPH method. Data are expressed as mean ± SD from three independent assays with five replicates each. Different letters indicate a significant difference (α < 0.05, Tukey’s multiple comparison test).

**Figure 4 ijms-26-08350-f004:**
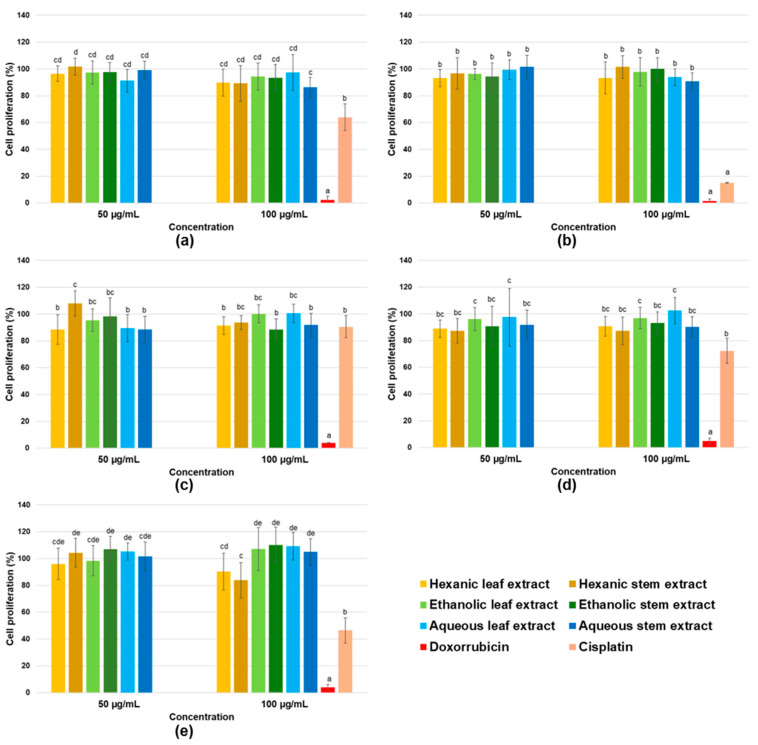
Cell proliferation percentages determined by the MTT assay for MDA-MB-231 invasive breast adenocarcinoma cells (**a**), HeLa cervical carcinoma cells (**b**), A549 lung carcinoma cells (**c**), HCT 116 colon carcinoma cells (**d**), and 22Rv1 prostate carcinoma cells (**e**) treated with crude hexane, ethanol, and water extracts from leaves and stems of *P. gracile*. Data are presented as mean ± SD from three independent assays with three replicates each. Different letters indicate significant differences (α < 0.05, Tukey’s multiple comparison test).

**Figure 5 ijms-26-08350-f005:**
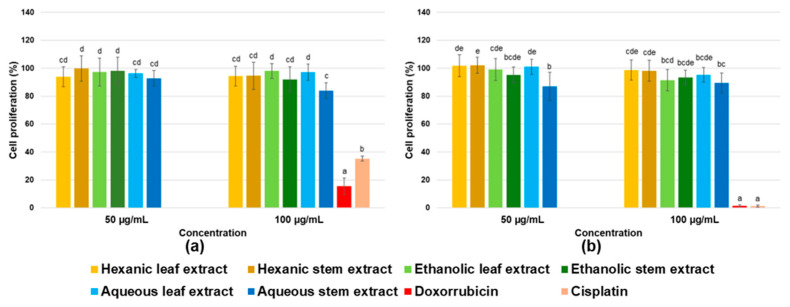
Cell proliferation percentages determined by MTT for non-cancerous ARPE-19 retinal epithelial cells (**a**) and hFOB 1.19 osteoblasts (**b**), treated with crude hexane, ethanol, and water extracts from leaves and stems of *P. gracile*. Data are presented as mean ± SD from three independent assays with three replicates each. Different letters indicate statistically significant difference (α < 0.05, Tukey’s multiple comparison test).

**Figure 6 ijms-26-08350-f006:**
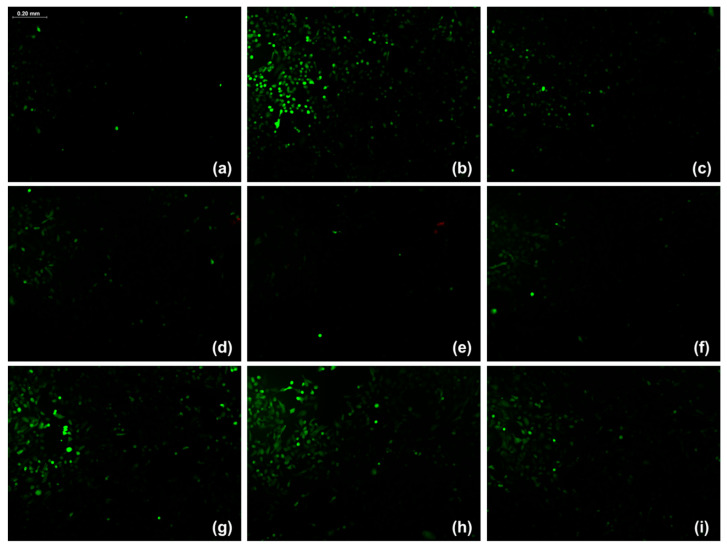
Fluorescence microscopy images (10X) of ARPE-19 cells subjected to doxorubicin-induced oxidative stress and treated with different *P. gracile* extracts. The fluorescence signal corresponds to the oxidation of DCFH-DA, indicating intracellular reactive oxygen species (ROS) generation. Panels: (**a**) untreated cells; (**b**) doxorubicin; (**c**) vitamin E + doxorubicin; (**d**) hexanic extract from leaf + doxorubicin; (**e**) hexanic extract from stem + doxorubicin; (**f**) ethanolic extract from stem + doxorubicin; (**g**) aqueous extract from leaf + doxorubicin; (**h**) aqueous extract from stem + doxorubicin; (**i**) ethanolic extract from leaf + doxorubicin. All images were acquired at the same magnification (10×).

**Table 1 ijms-26-08350-t001:** Profile of volatile compounds identified in leaves of *P. gracile* plant.

RT (min)	Compound	Area (%)
4.700	Hexanal	0.96 ± 0.09 ^fghi^
6.192	2-Hexenal	1.16 ± 0.21 ^ij^
7.915	Heptanal	0.28 ± 0.04 ^abcd^
9.059	α-Thujene	0.14 ± 0.01 ^ab^
9.332	α-Pinene	0.38 ± 0.03 ^abcde^
11.247	Sabinene	1.63 ± 0.28 ^jk^
12.410	β-Myrcene	35.20 ± 0.61 ^q^
12.928	Octanal	0.69 ± 0.02 ^defghi^
13.745	Fenchene	0.20 ± 0.01 ^abcd^
14.423	D-Limonene	2.02 ± 0.08 ^k^
14.858	trans-β-Ocimene	0.14 ± 0.01 ^ab^
15.482	cis-β-Ocimene	0.40 ± 0.02 ^abcde^
16.144	Carene	0.22 ± 0.02 ^abcd^
16.843	Sabinene hydrate	0.12 ± 0.04 ^ab^
17.010	1-Decanol	0.13 ± 0.03 ^ab^
17.926	Dihydrocarveol	0.09 ± 0.01 ^ab^
18.852	Linalool	3.42 ± 0.11 ^m^
19.142	Nonanal	1.64 ± 0.03 ^jk^
20.737	Terpinolene	0.09 ± 0.01 ^ab^
22.482	1,2-Epoxy-5,9-cyclododecadiene	0.17 ± 0.02 ^abc^
22.945	Z-Geraniol	0.04 ± 0.01 ^a^
23.618	1-Nonanol	0.22 ± 0.13 ^abcd^
24.416	β-Citral	0.11 ± 0.02 ^ab^
24.981	α-Terpineol	0.19 ± 0.03 ^abc^
25.952	Decanal	5.50 ± 0.10 ^n^
26.520	β-Cyclocitral	0.08 ± 0.00 ^ab^
27.345	Thymyl methyl ether	0.04 ± 0.01 ^a^
29.353	2-Decenal, (E)-	0.02 ± 0.00 ^a^
29.650	2-Decen-1-ol, (E)-	0.05 ± 0.01 ^ab^
29.925	Nonanoic acid	0.01 ± 0.01 ^a^
30.054	1-Dodecanol	0.19 ± 0.03 ^abcd^
30.324	Phytol	0.03 ± 0.00 ^a^
30.606	1-Octen-3-ol	0.47 ± 0.03 ^abcdef^
31.257	1-Pentadecene	1.17 ± 0.06 ^ij^
32.004	Tetradecanal	0.06 ± 0.00 ^ab^
33.216	Valencene	0.19 ± 0.01 ^abcd^
33.826	α-Guaiene	1.07 ± 0.01 ^hi^
34.017	α-Cubebene	1.01 ± 0.03 ^ghi^
34.159	Eugenol	0.01 ± 0.00 ^a^
35.014	Cyclosativene	0.29 ± 0.02 ^abcd^
35.449	Copaene	3.58 ± 0.10 ^m^
35.655	β-Maaliene	0.05 ± 0.00 ^a^
36.056	γ-Cadinene	0.91 ± 0.03 ^fghi^
36.157	β-Elemene	0.55 ± 0.04 ^bcdefg^
36.712	Cyperene	2.85 ± 0.09 ^l^
37.644	Caryophyllene	13.56 ± 0.45 ^p^
37.991	Isomethyl ionone	3.64 ± 0.20 ^m^
38.251	α-Bergamotene	0.28 ± 0.02 ^abcd^
38.429	Thujopsene	0.18 ± 0.02 ^abc^
38.899	γ-Muurolene	0.25 ± 0.02 ^abcd^
39.174	Humulene	0.82 ± 0.06 ^efghi^
39.804	2-Dodecenal, (E)-	0.25 ± 0.03 ^abcd^
40.056	Eremophilene	0.65 ± 0.06 ^cdefgh^
40.995	Germacrene B	0.37 ± 0.12 ^abcde^
41.167	α-Muurolene	0.16 ± 0.04 ^abc^
41.294	Longipinene	0.13 ± 0.04 ^ab^
41.535	Aromandendrene	0.23 ± 0.02 ^abcd^
41.983	α-Isomethyl ionone	6.07 ± 0.37 ^o^
42.464	Ledol	5.07 ± 0.61 ^n^
44.396	Spathulenol	0.13 ± 0.03 ^ab^
44.602	Caryophyllene oxide	0.27 ± 0.05 ^abcd^
48.199	Cadalene	0.14 ± 0.04 ^ab^

The results are expressed as the percentage of the total peak area corresponding to volatile compounds identified in *P. gracile* leaf samples (dry weight basis). Data are presented as mean ± SD, based on three independent analyses. Different superscript letters indicate significant differences in peak area values (α < 0.05, Tukey’s multiple comparison test).

**Table 2 ijms-26-08350-t002:** Profile of volatile compounds identified in stems of *P. gracile* plant.

RT (min)	Compound	Area (%)
6.568	Butanoic acid, 2-methyl-	0.36 ± 0.13 ^abc^
12.214	β-Myrcene	5.29 ± 1.86 ^fg^
14.387	D-Limonene	0.40± 0.14 ^abc^
18.786	Linalool	0.11 ± 0.05 ^a^
19.112	Nonanal	0.12 ± 0.06 ^a^
23.455	Ethanone, 1-(2-methyl-1-cyclopenten-1-yl)-	2.44 ± 0.62 ^de^
25.860	Decanal	0.10 ± 0.05 ^a^
26.478	β-Cyclocitral	0.02 ± 0.01 ^a^
29.529	2-Furanmethanol, tetrahydro-	0.03 ± 0.03 ^a^
30.532	Thymol	0.02 ± 0.01 ^a^
31.231	Dodec-1-ene	0.56 ± 0.27 ^abcd^
31.720	Thunbergol	0.02 ± 0.01 ^a^
31.975	Bicyclo [3.1.1]heptan-3-one, 2-(but-3-enyl)-6,6-dimethyl-	0.14 ± 0.16 ^ab^
32.746	Silphiperfol-5-ene	0.34 ± 0.09 ^abc^
33.197	β-Patchoulene	0.31 ± 0.09 ^abc^
33.352	Isogermacrene D	0.02 ± 0.00 ^a^
33.816	Silphinene	1.88 ± 0.33 ^abcd^
34.010	α-Cubebene	2.05 ± 0.29 ^bcde^
34.275	α-Guaiene	0.01 ± 0.00 ^a^
34.550	Bicyclo [2.2.1]heptane-2,5-dione, 1,7,7-trimethyl-	0.09 ± 0.10 ^a^
34.695	(±)-Cadinene	0.02 ± 0.01 ^a^
34.992	Cyclosativene	0.45 ± 0.11 ^abc^
35.458	Copaene	5.54 ± 0.05 ^fe^
35.641	β-Guaiene	0.09 ± 0.04 ^a^
35.777	α-Guaiene	0.07 ± 0.02 ^a^
36.047	β-Cubebene	1.56 ± 0.25 ^abcd^
36.155	β-Elemene	0.94 ± 0.12 ^abcd^
36.741	Cyperene	6.52 ± 0.59 ^g^
37.224	α-Patchoulene	0.64 ± 0.10 ^abcd^
37.697	Caryophyllene	19.07 ± 2.20 ^k^
38.041	Isomethyl ionone	9.57 ± 0.47 ^h^
38.247	α-Bergamotene, cis-	0.77 ± 0.06 ^abcd^
38.431	β-Bourbonene	0.48 ± 0.02 ^abc^
39.180	α-Humulene	2.12 ± 0.01 ^cde^
40.356	Germacrene D	3.94 ± 0.03 ^ed^
41.530	Aromandendrene	0.75 ± 0.06 ^a^
42.103	α-Isomethyl ionone	16.74 ± 1.57 ^j^
42.555	Kessane	14.00 ± 1.88 ^i^
42.746	Viridiflorol	0.24 ± 0.02 ^abc^
44.062	Thymyl 2-methylbutanoate	0.09 ± 0.01 ^a^
44.399	Spathulenol	0.59 ± 0.09 ^abcd^
44.604	Caryophyllene oxide	0.93 ± 0.13 ^abcd^
45.018	Salvial-4(14)-en-1-one	0.10 ± 0.02 ^a^
45.392	Carvacryl acetate	0.13 ± 0.06 ^a^
45.696	Ledol	0.07 ± 0.01 ^a^
46.794	Isoaromadendrene epoxide	0.10 ± 0.01 ^a^
47.546	Aromadendrene	0.13 ± 0.01 ^abcd^
48.658	(+)-Ledene	0.04 ± 0.01 ^a^

The results are expressed as the percentage of the total peak area corresponding to volatile compounds identified in *P. gracile* stem samples (dry weight basis). Data are presented as mean ± SD, based on three independent analyses. Different superscript letters indicate significant differences in peak area values (α < 0.05, Tukey’s multiple comparison test).

**Table 3 ijms-26-08350-t003:** Fatty acid profile identified in leaves and stems of *P. gracile* plant.

Standards Name	Name	mg/kg of Leaves	mg/kg of Stems
C4:0	Butyric acid	nd *	148.54 ± 23.21
C6:0	Caproic acid	nd	56.88 ± 7.25
C8:0	Caprylic acid	nd	52.66 ± 7.19
C10:0	Capric acid	nd	nd
C11:0	Hendecanoic acid	nd	nd
C12:0	Lauric acid	nd	63.91 ± 28.04
C14:0	Myristic acid	802.68 ± 68.18	620.47 ± 153.12
C15:0 iso		nd	70.69 ± 6.01
C15:0 anteiso		nd	nd
C14:1	Myristoleic acid	50.17 ± 64.63	63.33 ± 37.59
C15:0	Pentadecylic acid	1750.22 ± 103.95	1798.39 ± 518.99
C16:0 iso		421.69 ± 48.46	169.65 ± 72.66
C15:1	cis-10-pentadecenoic acid	nd	nd
C16:0	Palmitic acid	5978.89 ± 72.86	3734.71 ± 928.87
C17:0 iso		289.84 ± 21.40	nd
C16:1 c9/C16:1 n7	Palmitoleic acid	nd	nd
C17:0	Margaric acid	63.84 ± 45.70	73.33 ± 32.93
C16:3 n4	cis-6,9,12-hexadecatrienoic acid	nd	nd
C17:0 c9,10		nd	nd
C17:1 cis 10		nd	nd
C16:2 n4	cis-9, 12-hexadecanoic acid	nd	nd
C18:0	Stearic acid	600.03 ± 13.43	298.72 ± 129.71
C18:1 t9	Elaidic acid	nd	nd
C18:1 c9/C18:1 n9	Oleic acid	408.40 ± 1.69	379.36 ± 151.56
C18:1 n7/C18:1 c11		155.06 ± 42.26	nd
C19:0	Nonadecanoic acid	nd	nd
C18:2 t9,12		nd	nd
C19:0 c9,10		nd	nd
C18:2 c9,12/C18:2 n6	Linolelaidic acid	4855.88 ± 193.94	4607.57 ± 1188.75
C18:2 n4	Linoleic acid	nd	nd
C20:0	Arachidic acid	138.31 ± 9.49	328.02 ± 125.78
C18:3 c6,9,12 gamma/C18:3 n6	Gamma-linolenic acid	nd	nd
C18:3 n4	cis-9, 11,14-octadecatrienoic acid	nd	nd
C20:1 c11/C20:1 n9	Gadoleic acid	nd	nd
C18:3 c9,12,15 alpha/C18:3 n3	Alpha-linolenic acid	13619.49 ± 322.41	3692.29 ± 719.71
C21:0	Heneicosanoic acid	69.42 ± 29.22	nd
C18:4 n3	Stearidonic acid	nd	nd
C20:2 c11,14		nd	nd
C22:0	Behenic acid	541.38 ± 32.68	1000.64 ± 410.94
C20:3 c8,11,14		nd	nd
C22:1 n11		nd	nd
C22:1 c13/C22:1 n9		nd	nd
C20:3 c11,14,17		nd	nd
C20:4 c5,8,11,14/C20:4 n6		nd	nd
C23:0	Tricosanoic acid	242.80 ± 28.77	nd
C20:4 n3		nd	nd
C22:2 c13,16		nd	nd
C24:0	Lignoceric acid	452.52 ± 64.12	nd
C20:5 c5,8,11,14,17 EPA/C20:5 n3	Eicosapentaenoic acid	nd	nd
C24:1 c15/C24:1 n9		348.21 ± 53.42	58.14 ± 49.76
C24:2 n6	Tetracosatetraenoic acid	214.69 ± 80.33	nd
C21:5 n3	Heneicosapentaenoic acid	nd	nd
C22:5 n6	Osbond acid	nd	nd
C22:5 n3	Clupanodonic acid	nd	nd
C22:6 c4,7,10,13,16,19 DHA/C22:6 n3	Docosahexaenoic acid	nd	nd

The results are expressed as mg/kg of sample (leaves or stem) on a dry weight basis. Data are expressed as mean ± SD, based on two independent analyses. * nd: not detected (not present).

**Table 4 ijms-26-08350-t004:** The dry basis yield of crude extracts obtained from *P. gracile*.

Crude Extract	Yield (%)
Hexanic of leaves	3.25 ± 0.58 ^a^
Hexanic of stems	2.03 ± 0.59 ^a^
Ethanolic of leaves	5.92 ± 1.72 ^a^
Ethanolic of stems	3.8 ± 1.43 ^a^
Aqueous of leaves	14.45 ± 4.66 ^b^
Aqueous of stems	6.35 ± 1.20 ^a^

The results are expressed as percentage yield based on initial dry weight (20 g of *P. gracile*, either leaf or stem). Data are expressed as mean ± SD, based on four independent extractions. Different letters indicate a significant difference (α < 0.05, Tukey’s multiple comparison test).

**Table 5 ijms-26-08350-t005:** IC_50_ values for the ABTS radical of the crude extracts obtained from *P. gracile*.

Crude Extract	IC_50_ (µg/mL)
Hexanic of leaves	68.56 ± 19.20 ^b^
Hexanic of stems	42.71 ± 18.67 ^ab^
Ethanolic of leaves	23.08 ± 1.79 ^a^
Ethanolic of stems	16.49 ± 6.65 ^a^
Aqueous of leaves	19.15 g ± 6.80 ^a^
Aqueous of stems	38.94 ± 1.58 ^ab^

Data are presented as mean ± SD from three independent assays. Different letters indicate significant difference (α < 0.05, Tukey’s multiple comparison test).

**Table 6 ijms-26-08350-t006:** Results obtained from evaluation of crude extracts from *P. gracile* using the FRAP method.

Crude Extract	mmol TE/g of Extract
Hexanic of leaves	8.16 ± 1.42 ^a^
Hexanic of stems	12.00 ± 2.04 ^a^
Ethanolic of leaves	28.53 ± 1.61 ^b^
Ethanolic of stems	21.39 ± 4.56 ^b^
Aqueous of leaves	22.67 ± 3.40 ^b^
Aqueous of stems	9.28 ± 1.94 ^a^

Data are expressed as mean ± SD from one assay with three replicates each. Different letters indicate significant difference (α < 0.05, Tukey’s multiple comparison test).

**Table 7 ijms-26-08350-t007:** Relative fluorescence intensity of ARPE-19 cells treated with *P. gracile* extracts and exposed to doxorubicin-induced oxidative stress, as measured by the DCFH-DA assay.

Treatment (Concentration)	Relative Fluorescence Intensity (I/Io)
Untreated cells	1.00
Hexanic extract from leaves (100 µg/mL)	0.63 ± 0.19 ^abc^
Hexanic extract from stems (100 µg/mL)	0.42 ± 0.14 ^a^
Ethanolic extract from leaves (100 µg/mL)	0.75 ± 0.20 ^bcd^
Ethanolic extract from stems (100 µg/mL)	0.57 ± 0.22 ^ab^
Aqueous extract from leaves (100 µg/mL)	1.17 ± 0.21 ^e^
Aqueous extract from stems (100 µg/mL)	1.02 ± 0.20 ^de^
Vitamin E (0.4 mM)	0.90 ± 0.21 ^cde^

Data are presented as mean ± SD from three independent assays, each performed in triplicate. Different letters indicate significant differences (α < 0.05, Tukey’s multiple comparison test).

## Data Availability

The original contributions presented in this study are included in the article. Further inquiries can be directed to the corresponding authors.

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
