# Peer review of "Phytochemical Profile and Bioactive Evaluation of Porophyllum gracile"

_ijms, 2025, doi:10.3390/ijms26178350_

Round 1
Reviewer 1 Report
Comments and Suggestions for Authors
- Line 28: The abstract does not suggest any future research directions or practical applications of the findings.
- Replace the keywords inserted in the title.
- The introduction is too brief. Please provide more details about the importance of evaluating the phytochemical profile and bioactive evaluation of Porophyllum Gracile
- Line 54: Please add hypotheses before mentioning the objectives.
- In Table 1 and 2: Please add letters to the means to indicate statistical significance.
- The legend of figure 1 need enhancement.
- Line 276: Please clarify whether the Porophyllum gracile samples were collected from a single location or multiple locations. Also, please state the sample size used in this study
- In the discussion section, some information is repeated in the Results section. The discussion section is somewhat lacking in depth and could be strengthened. The authors should provide recommendations for future research directions that could build upon or address limitations of the current work.
- The conclusion is long, and it should be rewritten again to summarize the key findings.
- Most of the references used are old and need to be updated.
Author Response
Thank you very much for taking the time to review this manuscript. Below, you will find the answers to your comments, and in the attached file, the corresponding corrections highlighted in red. We would like to note that, in addition to the aforementioned, we have incorporated new results corresponding to the evaluation of cellular antioxidant activity.
Comment 1: Line 28: The abstract does not suggest any future research directions or practical applications of the findings.
Response: Thank you for your comment. We have modified the Abstract according to your recommendations, incorporating suggestions for future research directions and potential practical applications of the findings, in addition to adding the new results obtained.
Location in the manuscript: page 1, paragraph 1, line 15.
Comment 2: Replace the keywords inserted in the title.
Response: Thank you for your suggestion. We have replaced the keywords found in the title with “Bioactive Compounds.”
Comment 3: The introduction is too brief. Please provide more details about the importance of evaluating the phytochemical profile and bioactive evaluation of Porophyllum gracile.
Response: Thank you for your comment. We have expanded the Introduction section, addressing comments 3 and 4, to highlight the importance of evaluating the phytochemical profile and bioactive potential of Porophyllum gracile. The revised text emphasizes the knowledge gap regarding this species and underscores how a detailed phytochemical characterization can help identify previously unreported compounds, including bioactive molecules with potential applications. Additionally, we have incorporated a working hypothesis stating that P. gracile contains a high diversity of molecules with different chemical natures and high bioactive potential, particularly in vitro antioxidant and antiproliferative effects.
Location in the manuscript for comment 3: page 2, paragraph 3, line 56.
Location in the manuscript for comment 4: page 2, paragraph 4, line 61.
Comment 4: Line 54: Please add hypotheses before mentioning the objectives.
Response: Thank you for your comment. We have expanded the Introduction section, addressing comments 3 and 4, to highlight the importance of evaluating the phytochemical profile and bioactive potential of Porophyllum gracile. The revised text emphasizes the knowledge gap regarding this species and underscores how a detailed phytochemical characterization can help identify previously unreported compounds, including bioactive molecules with potential applications. Additionally, we have incorporated a working hypothesis stating that P. gracile contains a high diversity of molecules with different chemical natures and high bioactive potential, particularly in vitro antioxidant and antiproliferative effects.
Location in the manuscript for comment 3: page 2, paragraph 3, line 56.
Location in the manuscript for comment 4: page 2, paragraph 4, line 61.
Comment 5: In Table 1 and 2: Please add letters to the means to indicate statistical significance.
Response: Thank you for pointing this out. We have conducted the statistical analysis of the data and added letters indicating significant differences to Tables 1 and 2.
Location in the manuscript: page 2, Table 1, line 82; and page 4, Table 2, line 86.
Comment 6: The legend of Figure 1 needs enhancement.
Response: We agree with this comment; therefore, we have rewritten the legend of Figure 1.
Location in the manuscript: page 5, legend of Figure 1, line 103.
Comment 7: Line 276: Please clarify whether the Porophyllum gracile samples were collected from a single location or multiple locations. Also, please state the sample size used in this study.
Response: Thank you for pointing this out. We have modified section 4.1. Plant Material to clarify this point. The plant material was collected from a single location (Puerto de Yavaros, Sonora, Mexico), with an approximate total mass of 4 kg collected.
Location in the manuscript: page 14, paragraph 8, lines 337 and 341.
Comment 8: In the discussion section, some information is repeated in the Results section. The discussion section is somewhat lacking in depth and could be strengthened. The authors should provide recommendations for future research directions that could build upon or address limitations of the current work.
Response: We agree with the comment and have made several changes to the Discussion section to avoid repetition with the Results section, enhance the depth of the discussion, and include recommendations for future research directions addressing the limitations of the current study.
Location in the manuscript: page 14, paragraphs 4–7.
Comment 9: The conclusion is long, and it should be rewritten again to summarize the key findings.
Response: We appreciate the reviewer’s observation. In response, the Conclusions section has been rewritten to provide a more concise summary of the key findings.
Location in the manuscript: page 18, paragraph 3, line 498.
Comment 10: Most of the references used are old and need to be updated.
Response: We appreciate this comment; therefore, we have updated the relevant references throughout the text, excluding the articles related to the genus Porophyllum and the methodologies on which the study was based. As a result of this change, the order of all references has been adjusted.

Reviewer 2 Report
Comments and Suggestions for Authors
In the manuscript “Phytochemical profile and bioactive evaluation of Porophyllum gracile” by Ruiz-Almada et al., the authors report the chemical characterization of extracts from leaves and stems of P. gracile and evaluate some of the extracts' bioactive properties, including antiproliferative activity against cancer and non-cancer cell lines and in vitro antioxidant activity.
Overall, the manuscript is well-structured. The introduction is comprehensive, and the results are clearly presented and well discussed.
The strength of this manuscript lies in the fact that it is the first study to provide a thorough analysis of the chemical composition of P. gracile extracts. The literature only reports a single study from 1998, which describes a very partial chemical characterization of extracts obtained using different methods.
On the other hand, the weakness lies in determining the biological properties, which currently provides very limited insights.
Based on the results of the MTT assay, the extracts did not show significant activity on either cancerous or non-cancerous cells, suggesting a likely absence of antiproliferative effects. Regarding antioxidant activity, three different chemical assays were used to assess the total antioxidant capacity or scavenging ability against various radical species; however, these assays do not provide information on potential cellular targets.
Therefore, I believe that the manuscript should be complemented with in vitro assays using cellular models to determine a possible role of the extracts in protecting against oxidative stress, for example, by measuring intracellular ROS levels or evaluating changes in the expression of key proteins involved in the cell’s antioxidant response.
Minor revisions:
- Page 3, line 73 and page 4, line 77: P. gracile should be written in italics.
- Table 4: It is not clearly stated—nor specified in the Materials and Methods section—how the yield of the crude extracts is expressed (a percentage of what? The initial dry weight?).
- Page 7, line 126: “According to literature, there are IC50 value are used to classify…..” There appears to be a typographical error, as the sentence is unclear or does not read correctly.
- Figures 2,3,4, and 5: The labels below the graphs indicating the concentrations and the different samples used are difficult to read in the printed version. The font size should be increased, as far as editorial constraints allow.
Author Response
Comment 1: Page 3, line 73 and page 4, line 77: P. gracile should be written in italics.
Response: Thank you for pointing this out. We agree with this comment. Therefore, we have italicized P. gracile.
Location in the manuscript: page 4, legend of Table 1, line 84; and page 5, legend of Table 2, line 88.
Comment 2: Table 4: It is not clearly stated—nor specified in the Materials and Methods section—how the yield of the crude extracts is expressed (a percentage of what? The initial dry weight?).
Response: Thank you for pointing this out. We agree with this comment. Therefore, we have modified the legend of Table 4 to clarify this information.
Location in the manuscript: page 7, legend of Table 4, line 123.
Comment 3: Page 7, line 126: “According to literature, there are IC50 value are used to classify…..” There appears to be a typographical error, as the sentence is unclear or does not read correctly.
Response: Thank you for pointing this out. We agree with this comment. Therefore, we have corrected the error and replaced the sentence with: “According to the literature, IC50 values are used to classify the antioxidant potential of extracts as follows:”.
Location in the manuscript: page 8, paragraph 2, line 138.
Comment 4: Figures 2, 3, 4, and 5: The labels below the graphs indicating the concentrations and the different samples used are difficult to read in the printed version. The font size should be increased, as far as editorial constraints allow.
Response: Thank you for pointing this out. We agree with this comment. Therefore, we have increased the font size in Figures 2, 3, 4, and 5.
Location in the manuscript: page 8, Figure 2, line 131; page 9, Figure 3, line 156; page 10, Figure 4, line 175; and page 10, Figure 5, line 185.
Additional Observation (no numbered comment): Based on the results of the MTT assay, the extracts did not show significant activity on either cancerous or non-cancerous cells, suggesting a likely absence of antiproliferative effects. Regarding antioxidant activity, three different chemical assays were used to assess the total antioxidant capacity or scavenging ability against various radical species; however, these assays do not provide information on potential cellular targets. Therefore, it was suggested that the manuscript be complemented with in vitro assays using cellular models to determine a possible role of the extracts in protecting against oxidative stress, for example, by measuring intracellular ROS levels or evaluating changes in the expression of key proteins involved in the cell’s antioxidant response.
Response: We appreciate the reviewer’s valuable observation and fully agree with the comment. Accordingly, we have incorporated an additional section in the manuscript, in which the antioxidant activity was evaluated in the non-cancerous ARPE-19 cell line. We believe that this information complements the study and strengthens the in vitro results by providing further insight into the biological relevance of the antioxidant activity observed.

Round 2
Reviewer 1 Report
Comments and Suggestions for Authors
The revised version is acceptable.